# Novel Concepts of Treatment for Patients with Myelofibrosis and Related Neoplasms

**DOI:** 10.3390/cancers12102891

**Published:** 2020-10-09

**Authors:** Prithviraj Bose, Lucia Masarova, Srdan Verstovsek

**Affiliations:** Department of Leukemia, University of Texas MD Anderson Cancer Center, Houston, TX 77030, USA; lmasarova@mdanderson.org (L.M.); sverstov@mdanderson.org (S.V.)

**Keywords:** imetelstat, PRM-151, KRT-232, CPI-0610, ropeginterferon alfa-2b

## Abstract

**Simple Summary:**

Myelofibrosis (MF) is an advanced form of a group of rare, related bone marrow cancers termed myeloproliferative neoplasms (MPNs). Some patients develop myelofibrosis from the outset, while in others, it occurs as a complication of the more indolent MPNs, polycythemia vera (PV) or essential thrombocythemia (ET). Patients with PV or ET who require drug treatment are typically treated with the chemotherapy drug hydroxyurea, while in MF, the targeted therapies termed Janus kinase (JAK) inhibitors form the mainstay of treatment. However, these and other drugs (e.g., interferons) have important limitations. No drug has been shown to reliably prevent the progression of PV or ET to MF or transformation of MPNs to acute myeloid leukemia. In PV, it is not conclusively known if JAK inhibitors reduce the risk of blood clots, and in MF, these drugs do not improve low blood counts. New approaches to treating MF and related MPNs are, therefore, necessary.

**Abstract:**

Janus kinase (JAK) inhibition forms the cornerstone of the treatment of myelofibrosis (MF), and the JAK inhibitor ruxolitinib is often used as a second-line agent in patients with polycythemia vera (PV) who fail hydroxyurea (HU). In addition, ruxolitinib continues to be studied in patients with essential thrombocythemia (ET). The benefits of JAK inhibition in terms of splenomegaly and symptoms in patients with MF are undeniable, and ruxolitinib prolongs the survival of persons with higher risk MF. Despite this, however, “disease-modifying” effects of JAK inhibitors in MF, i.e., bone marrow fibrosis and mutant allele burden reduction, are limited. Similarly, in HU-resistant/intolerant PV, while ruxolitinib provides excellent control of the hematocrit, symptoms and splenomegaly, reduction in the rate of thromboembolic events has not been convincingly demonstrated. Furthermore, JAK inhibitors do not prevent disease evolution to MF or acute myeloid leukemia (AML). Frontline cytoreductive therapy for PV generally comprises HU and interferons, which have their own limitations. Numerous novel agents, representing diverse mechanisms of action, are in development for the treatment of these three classic myeloproliferative neoplasms (MPNs). JAK inhibitor-based combinations, all of which are currently under study for MF, have been covered elsewhere in this issue. In this article, we focus on agents that have been studied as monotherapy in patients with MF, generally after JAK inhibitor resistance/intolerance, as well as several novel compounds in development for PV/ET.

## 1. Introduction

Given the universal activation of the Janus kinase (JAK) signal transducer and activator of transcription (STAT) signaling observed in the classic, Philadelphia chromosome (Ph)-negative myeloproliferative neoplasms (MPNs) [1], the central role that JAK inhibitors play in these diseases is not surprising [2,3]. In myelofibrosis (MF), ruxolitinib and fedratinib provide marked benefits to patients in terms of reduction of splenomegaly and improvement in symptoms [4,5,6]. Neither agent significantly ameliorates cytopenias, however, and data are lacking to support the use of either agent in patients with platelets <50 × 10^9^/L; indeed, anemia and thrombocytopenia are frequent adverse events (AEs), particularly in the first 12-24 weeks of therapy. Ruxolitinib, for which longer follow-up is available, is associated with a survival advantage in patients with intermediate-2/high-risk MF [7]. However, the effects of JAK inhibitors on the underlying bone marrow fibrosis and driver mutation allele burden are relatively modest [8,9], and these agents do not prevent transformation to blast phase (BP), a devastating complication. Survival has been reported by several groups to be dismal for patients who discontinue ruxolitinib [10,11,12]. Numerous drug development efforts are underway to address these unmet needs. Although derailed in earlier phase 3 trials which had mixed results [13,14,15,16], the JAK inhibitors momelotinib (which may improve anemia via suppression of hepatic hepcidin production [17]) and pacritinib (which is less myelosuppressive and may be more efficacious in patients with a “myelodepletive” phenotype [18]) have re-entered phase 3 trials in symptomatic, anemic patients post ruxolitinib and severely thrombocytopenic patients with MF, respectively (NCT04173494, NCT03165734). A plethora of novel agents are being studied in combination with ruxolitinib, either from the outset, or in “add on” fashion in patients with a suboptimal response to ruxolitinib; these are discussed in detail by Kuykendall et al. in this issue [19]. Others have been studied as single agents, mostly in the ruxolitinib “failure” setting, and are discussed below. Some of these have also been tested in combination with ruxolitinib. 

Hydroxyurea (HU) is by far the most commonly used cytoreductive drug in patients with polycythemia vera (PV), although pegylated interferon alfa is a reasonable alternative [20]. Very recently, ropeginterferon alfa-2b (a long-acting, monopegylated interferon) has been approved in Europe for patients with PV without symptomatic splenomegaly [21]. Options after failure of HU include ruxolitinib [22,23] and pegylated interferon alfa [24]. Ruxolitinib has not been shown to statistically significantly reduce the risk of thromboembolic events, a major goal of therapy, in patients with PV [25], although it does provide sustained control of the hematocrit, leukocyte and platelet counts, splenomegaly and symptoms [26]. While HU does reduce the risk of thromboembolic events compared to no cytoreductive therapy in patients with high-risk PV [27], it appears to fail in the splanchnic venous district [28]. Importantly, no therapy reliably reduces the risk of progression to MF or transformation to acute myeloid leukemia (AML). These observations have provided the impetus for exploration of a number of novel drug classes in patients with PV, discussed below. 

Drug development in essential thrombocythemia (ET) is difficult because of the long natural history of the disease, in which survival may not differ significantly from that of an age- and sex-matched healthy population [29]. Furthermore, there does not appear to be a correlation between the platelet count and thrombotic risk in ET [30,31,32]. Like for PV, both HU and pegylated interferon alfa are reasonable frontline options for cytoreductive therapy in ET; as in PV, HU is by far the more commonly used [33]. Anagrelide is used as a first-line treatment in some countries [34], but is generally utilized as a second-line agent in the US. Some experts use busulphan as well [35]. Ruxolitinib was studied in patients with HU-resistant/intolerant ET in a randomized trial, and failed to show benefits over best available therapy (BAT) in terms of response rates or rates of thrombosis, hemorrhage and transformation, but did improve symptoms to a greater extent than BAT [36]. However, ruxolitinib did provide long-term count and symptom control in a separate, open-label study [37], and continues to be developed for ET (NCT02577926, NCT02962388). Some of the novel agents that have shown efficacy in MF or PV and are discussed below have also been or are planned to be studied in patients with ET.

Figure 1 provides a timeline of major milestones in the pathogenesis and therapy of the three classic MPNs. Figure 2 depicts some of the cellular pathways that have been or can be targeted for therapeutic benefit in these diseases.

## 2. Telomerase Inhibition with Imetelstat in MF

In a pilot study in 33 patients with intermediate-2/high-risk MF, nearly half of whom had received prior JAK inhibitor therapy, the telomerase inhibitor imetelstat produced complete or partial responses in seven (21%), reversing bone marrow fibrosis in all four patients who achieved a complete response (CR) [38]. Three of these four patients also had molecular responses. Responses were restricted to *JAK2*-mutated and *ASXL1* wild type patients, and the CR rate was significantly higher in patients with *SF3B1* or *U2AF1* mutations. Myelosuppression and liver enzyme elevation were common. Subsequent mechanistic studies showed that imetelstat can selectively deplete MF stem and progenitor cells via dose-dependent inhibition of telomerase activity and induction of apoptosis [39]. 

These findings led to a larger, multi-center trial (IMBARK™) of two doses of imetelstat, 4.7 and 9.4 mg/kg, administered intravenously every 3 weeks, in 107 patients with intermediate-2/high-risk MF who had relapsed after, or whose disease was refractory to, prior JAK inhibitor treatment, as defined in Table 1. Forty-eight patients were enrolled on the 4.7 mg/kg arm, which was subsequently closed due to insufficient activity and patients still receiving treatment crossed over to the 9.4 mg/kg arm. In the higher dose arm (n = 59), the rate of ≥35% spleen volume reduction (SVR) at week 24 was 10.2%, and that of ≥50% reduction in total symptom score (TSS) was 32.2% [40]. Remarkably, median overall survival (OS) was 19.9 months in the 4.7 mg/kg arm and 28.1 months in the 9.4 mg/kg arm. Rates of 12-week transfusion independence (TI), bone marrow fibrosis reduction and ≥25% reduction in driver mutation allele burden in the 9.4 mg/kg arm were 25%, 43.2% and 42.1%, respectively. Spleen and symptom responses, as well as OS, correlated with achievement of ≥50% reduction in telomerase activity and in human telomerase reverse transcriptase (hTERT) expression level [41]. Short telomere length at baseline, as well as higher baseline hTERT expression level, also predicted improved clinical outcomes and longer OS (trend only) in the 9.4 mg/kg arm. Finally, 25% of participants in the IMBARK™ trial had triple-negative disease, a well-established poor-prognosis subset in primary myelofibrosis (PMF) [42,43]. At the 9.4 mg/kg dose, triple-negative patients had higher rates of ≥35% SVR (18.8% versus 7.3%) and ≥50% TSS reduction (50% versus 24.4%) at 24 weeks than non-triple-negative patients, as well as superior OS (median, 35.9 months versus 24.6 months, *p* = 0.05) [44]. Despite 92% of triple-negative patients having grade 3 bone marrow fibrosis, the rate of bone marrow fibrosis improvement was higher (50%) than in non-triple-negative patients (39.1%). Importantly, triple-negative patients on the 9.4 mg/kg arm had shorter baseline telomere length and higher baseline hTERT expression compared to non-triple-negative patients. Imetelstat has also been studied in ET, with high reported efficacy in terms of hematologic and molecular response rates [45], but current development efforts are focused on MF. 

## 3. Targeting Bone Marrow Fibrosis in MF

Verstovsek et al. demonstrated that, contrary to popular belief, the fibrosis-engendering fibrocytes in the bone marrow of individuals with PMF are clonal (neoplastic, i.e., carry the driver mutation) and derived from monocytes [46]. Japanese investigators validated and extended this work to show that fibrocyte differentiation is triggered by myeloproliferative leukemia (MPL) receptor activation (by thrombopoietin) and that circulating monocytes highly expressing signaling lymphocytic activation molecule F7 (SLAMF7) were possible fibrocyte precursors [47]. They went on to show that particularly in *JAK2*-mutated MF patients, the circulating SLAMF7^high^ monocyte percentage was significantly elevated, and correlated with *JAK2* V617F allele burden [48]. While the former work from our group served as the basis for studying PRM-151 (recombinant pentraxin-2, or serum amyloid protein) in patients with MF (discussed below), the Japanese group identified the anti-SLAMF7 monoclonal antibody, elotuzumab, as a therapeutic candidate in MF. Elotuzumab inhibited fibrocyte differentiation in vitro and ameliorated bone marrow fibrosis and splenomegaly induced by romiplostim (thrombopoietin agonist) administration in humanized mice [48]. Accordingly, there are plans to study this agent in patients with *JAK2*-mutated MF who are not candidates for JAK inhibitor therapy (NCT04517851). Other groups have identified Gli1^+^ mesenchymal stromal cells as possible precursors of myofibroblasts, leading to interest in targeting these cells with Gli antagonists [49]. Finally, transforming growth factor beta (TGF-β) has long been implicated in the pathogenesis of bone marrow fibrosis, although it is difficult to therapeutically target [50]. It was recently demonstrated that TGF-β1, which is produced by hematopoietic cells, including fibrocytes, promotes the differentiation of neoplastic monocytes to fibrocytes in *JAK2* V617F-induced PMF, and that elevated plasma TGF-β1 levels can be returned to normal by monocyte depletion [51]. AVID200 is a novel, potent and highly specific TGF-β1/3 “trap” that is currently being studied in a phase 1 trial in patients with MF (NCT03895112). 

Serum amyloid P or pentraxin-2 is an endogenous plasma protein that functions as a soluble pattern recognition receptor of the innate immune system and may localize specifically to sites of injury to aid in removal of damaged tissue [52]. PRM-151 was studied, both alone and in combination with ruxolitinib, in 27 patients with MF, 18 of whom went on to an open-label extension (OLE) phase after the initial six cycles [53]. These 18 patients were on-study for 30.9 months (median), and derived clinical benefits in terms of reduction of splenomegaly and symptoms, irrespective of concomitant ruxolitinib use, as well as improvements in anemia and thrombocytopenia. PRM-151 was extremely well tolerated, and nine patients (50%) had reticulin grade improvements, with eight patients (44%) having collagen grade improvements. Three doses of PRM-151, 0.3 mg/kg, 3 mg/kg and 10 mg/kg, were then studied in 97 patients with MF who were ineligible for, intolerant of, or had responded inadequately to ruxolitinib [54]. PRM-151 was administered intravenously on days 1, 3 and 5 in the first 28-day cycle, and then only on day 1 for a total of nine cycles, after which patients could go on to an OLE phase. Discontinuation rates were high, with only 51 patients (53%) completing nine cycles, 48 of whom (49% overall) went on to the OLE phase. Seventy-four patients (76%) had received prior ruxolitinib, 84% had baseline hemoglobin <10 g/dL and 59% had baseline platelets of <50 x 10^9^/L. Twenty-seven (27.8%) patients experienced decreases in bone marrow fibrosis grade at any time; this was true for collagen grade as well, with some patients having two-grade improvements. Five of 31 patients (16%) achieved red blood cell (RBC) TI, while platelet TI was achieved in six of 13 patients (46%). Hemoglobin, but not platelet, improvement appeared to correlate with bone marrow fibrosis grade reduction. The rate of ≥35% SVR was not reported, but 32 of 94 (34%) evaluable patients had a ≥50% reduction in TSS at any time. Benefits appeared more pronounced in the 10 mg/kg dosing group, despite poorer baseline characteristics in this group. 

Yet other approaches to target bone marrow fibrosis are being pursued. Although simtuzumab, a monoclonal antibody against the extracellular matrix enzyme lysyl oxidase-like 2 (LOXL2), failed to show clinical benefit at 24 weeks in a phase 2 study both with and without ruxolitinib [55], there is continued interest in pursuing this target with small-molecule approaches (NCT04054245). Alisertib, a small-molecule inhibitor of aurora kinase A, promoted polyploidization and differentiation of megakaryocytes with PMF-associated mutations and exhibited potent anti-fibrotic and anti-tumor activity in vivo in mouse models of PMF [56]. Clinical results with this agent were modest, however, in a phase 1 trial in 24 patients with MF either resistant/intolerant to (15) or ineligible for (9) JAK inhibitor therapy [57]. Spleen (by palpation), symptom and anemia responses (by the 2013 International Working Group for MPN Research and Treatment (IWG-MRT) criteria [58]) occurred in 29%, 32% and 11% of eligible patients, respectively. Improvement in megakaryocyte morphology and restoration of GATA1 staining were observed in most patients from whom serial bone marrow samples were available. GATA1 down-regulation related to a RPS14-deficient gene signature that is associated with defective ribosomal protein function has been reported to underlie megakaryocytic proliferation and atypia in PMF [59,60]. Inhibition of TGF-β1 signaling has also been shown to reverse the GATA1 (low) phenotype in mouse models [61]. 

## 4. Activating p53: MDM2 Inhibition in PV and MF

Loss of function mutations in *TP53* are uncommon in chronic phase MPNs [62,63], unlike in blast phase [64,65], and *JAK2* V617F induces overexpression of murine double minute 2 (MDM2), the physiologic negative regulator of p53 [66]. Inducing p53-dependent apoptosis via blockade of MDM2 is, therefore, an attractive concept. The ability of idasanutlin, both alone and in combination with pegylated interferon alfa, to target MPN stem and progenitor cells, has been shown in preclinical studies [67]. These observations formed the basis of a phase 1 trial of idasanutlin, given orally at a dose of 100 mg or 150 mg daily for 5 days per cycle, in 12 patients with high-risk, *JAK2* V617F^+^ PV (n = 11) or ET (n = 1) who had received at least one prior cytoreductive therapy and had been found to be resistant or intolerant [68]. Baseline MDM2 levels were higher in study participants than in normal controls, and plasma MIC-1 levels were significantly increased in PV patients following treatment with idasanutlin, providing evidence for p53 activation. Anti-emetic prophylaxis was routine, and no grade 3/4 gastrointestinal AEs occurred. The overall response rate (ORR) per the 2013 European LeukemiaNet (ELN)-IWG criteria [69] in part A (idasanutlin monotherapy) was 58% (three partial responses (PRs) and four CRs). Four non-responders from part A continued on to part B (pegylated interferon alfa added to idasanutlin); two of these patients responded (one PR and one CR) for a composite ORR of 75% (nine of 12 patients). All but one patient (with an inactivating *TP53* mutation) experienced symptom improvement. The median duration of response (DOR) was 16.8 months. These encouraging findings led to a global, multi-center, single arm, phase 2 trial of idasanutlin in HU-resistant/intolerant PV (NCT03287245). Regrettably, this study had to be terminated by the sponsor due to difficulties in managing the gastrointestinal toxicities in the setting of a multi-national, multi-center trial (personal communication, John Mascarenhas, MD). 

Another MDM2 inhibitor, KRT-232, is being studied in patients with *TP53* wild type MF that has relapsed after or is refractory to JAK inhibitor therapy (NCT03662126), as defined in Table 2 [70]. Importantly, this ongoing study does not enroll patients who are intolerant to JAK inhibitor therapy. A minimum platelet count of 50 x 10^9^/L is required. Four doses/schedules have been studied: 120 mg/d on days 1-7 every 3 weeks, 240 mg/d on days 1-7 every 3 or 4 weeks and 240 mg/day on days 1-5 every 4 weeks, with data on the first three cohorts (n = 82) presented at the 25^th^ Congress of the European Hematology Association (EHA) earlier this year. The second cohort, i.e., 240 mg/d on days 1-7 every 3 weeks, was closed to enrollment due to excessive toxicity. The 240 mg daily dose on days 1-7 of a 4-week cycle was found to be the most effective, with a best rate (at any time point) of ≥35% SVR of 16% (n = 25). The best rate of ≥50% TSS reduction was 30% at this dose and schedule (n = 27). Grade 3/4 treatment-emergent AEs consisted primarily of myelosuppression (predominantly anemia and thrombocytopenia) and gastrointestinal toxicity (mainly diarrhea). KRT-232 has received fast-track designation from the Food and Drug Administration (FDA) for MF relapsed after or refractory to JAK inhibitor therapy and will be studied against BAT in a global, randomized, phase 3 trial at a dose of 240 mg/d administered orally on days 1-7 every 4 weeks.

## 5. Novel epigenetic therapies for MF

Inhibitors of DNA methyltransferases have limited single-agent activity in MF [71], and histone deacetylase inhibitors (HDACis), while active, are difficult to tolerate long-term because of chronic, low-grade toxicities, and disease-modifying effects are slow to appear [72,73,74]. In recent years, the focus has shifted to the exploration of novel epigenetic targets for therapy of MF. 

Bromodomain and extra-terminal (BET) proteins are epigenetic “readers” that control the transcription of multiple oncoproteins of importance in MF, e.g., nuclear factor kappa B (NF-κB), c-Myc and B-cell lymphoma-2 (BCL-2). The combination of JAK and BET inhibition has been shown to be synergistic in preclinical models of MPN, both in vitro and in vivo [75]. Promising results have been reported with the combination of ruxolitinib and CPI-0610, both in JAK inhibitor-naïve patients with MF [76], and in those on ruxolitinib alone for ≥6 months with a “suboptimal” response, who received CPI-0610 as “add on” therapy [77]. These results from the ongoing MANIFEST trial (NCT02158858) have been discussed in detail by Kuykendall et al. and, indeed, the developmental path forward for this agent is in combination with ruxolitinib in the frontline setting (MANIFEST-2). However, data from the monotherapy cohort of the MANIFEST trial suggest that CPI-0610 can be a useful drug for anemia in some MF patients [78]. Among 43 patients with MF (16 RBC transfusion-dependent (TD) and 27 not) who were refractory to, intolerant of or ineligible for JAK inhibitor treatment, three of 14 TD patients (21.4%) achieved TI, and 11 of 19 non-TD patients (57.9%) with baseline hemoglobin <10 g/dL achieved a sustained ≥1.5 g/dL improvement in hemoglobin without RBC transfusions. The rates of ≥35% SVR (0%) and ≥50% TSS reduction (8.3%) at 24 weeks with CPI-0610 monotherapy in this cohort were, however, disappointing. 

Another novel epigenetic target in MPNs is the “eraser” enzyme, lysine-specific (histone) demethylase-1 (LSD1) [79]. The LSD1 inhibitor bomedemstat normalized or improved blood counts, reduced splenomegaly, bone marrow fibrosis and mutant allele burden, restored normal splenic architecture and improved survival in mouse models of MPN [80]. This agent is being studied in a phase 2 trial in intermediate-2/high-risk MF patients who are resistant to or intolerant of ruxolitinib (NCT03136185) [81]. A minimum baseline platelet count of 100 x 10^9^/L is required. As of the most recent update presented at the 2020 EHA Congress, 18 patients had been treated in the dose-finding phase 2a portion and 18 in the phase 2b portion. Dose-limiting toxicities (DLTs) were not observed and a maximum tolerated dose (MTD) was not identified. Bomedemstat dosing was up-titrated to a target platelet count of 50-100 x 10^9^/L in phase 2a and 50-75 × 10^9^/L in phase 2b. While most evaluable patients (i.e., those who had reached 24 weeks) experienced some SVR, symptom improvement and stable to improved hemoglobin, only one patient (8.3%) had a ≥35% SVR at 24 weeks, though several experienced ≥50% TSS reduction. A third of patients experienced dysgeusia. Bomedemstat has received fast-track designation from the FDA for both MF and ET, and is also being studied in patients with ET who have failed at least one standard therapy (NCT04254978).

Yet another epigenetic target, a “writer” enzyme, of interest in the MPNs is the arginine methyltransferase PRMT5. PRMT5 is overexpressed in primary MPN cells and PRMT5 inhibition has been shown to reverse the MPN phenotype in vivo in mouse models of both *JAK2* V617F- and *MPL* W515L-mediated disease [82]. The PRMT5 inhibitor PRT543 is being studied in a phase 1 trial in advanced solid tumors and hematologic malignancies, including relapsed/refractory MF (NCT03886831). 

## 6. Targeting the Anti-Apoptotic Machinery in MF

Preclinical evidence of synergism in MPN models between ruxolitinib and the BH3-mimetic ABT-737, which antagonizes both BCL-2 and B-cell lymphoma extra long (BCL-xL) [83], was successfully translated into a clinical trial of navitoclax, the clinical counterpart of ABT-737, added to ruxolitinib in patients with a suboptimal response to the latter [84], as discussed by Kuykendall et al. The combination of ruxolitinib and navitoclax will now be compared in two phase 3 trials, TRANSFORM-1 and -2, against ruxolitinib plus placebo in JAK2 inhibitor-naïve patients (NCT04472598) and against BAT in patients with disease relapsed after or refractory to JAK2 inhibitor therapy (NCT04468984). An obvious challenge with navitoclax, particularly when used in conjunction with ruxolitinib, is thrombocytopenia resulting from on-target inhibition of Bcl-xL [85]. Indeed, this phenomenon, seen in all the early clinical trials of navitoclax [86,87,88,89], thwarted its development and led to the reverse engineering of navitoclax to develop venetoclax, which is far more selective for Bcl-2 [90]. However, Bcl-xL is likely the more important target in the context of MPNs, at least *JAK2*-mutated MPNs [91,92]. A minimum baseline platelet count of ≥100 × 10^9^/L is required in the current “add on” trial of navitoclax in patients with a sub-optimal response to ruxolitinib (NCT03222609), and the dose is carefully up-titrated with initially weekly platelet count monitoring [84]. For this reason, it is difficult to envision a role for navitoclax in patients who are intolerant of ruxolitinib, as most toxicities associated with ruxolitinib are hematologic. Similar considerations would likely also apply to any future combination of navitoclax with fedratinib. Navitoclax is also being studied in a phase 1 trial (NCT04041050) in patients with ET and PV. 

Another novel approach targeting a different arm of the anti-apoptotic machinery involves the use of second mitochondrial activator of caspases (Smac) mimetics, also known as inhibitor of apoptosis (IAP) antagonists. Preclinical support for this strategy comes from studies that show that the MF milieu is rich in tumor necrosis factor alpha (TNF-α) and that *JAK2* V617F both promotes this and confers resistance to TNF-α, thereby encouraging clonal selection [93]. More recent work has revealed down-regulation of X-linked inhibitor of apoptosis (XIAP) and up-regulation of cellular IAP proteins 1 and 2 (cIAP1 and cIAP2) in MF CD34^+^ cells, contributing to greater TNF-α-induced NF-κB activation compared to normal bone marrow CD34^+^ cells [94]. These observations led to an investigator-initiated trial of LCL-161, an oral Smac mimetic, administered weekly in 50 patients with MF who had failed or were not eligible for JAK inhibitor therapy [95]. There was no minimum platelet count for eligibility. The ORR by the 2013 IWG-MRT criteria was 30%, and consisted of clinical improvement (CI) in spleen, symptoms and anemia, as well as a complete cytogenetic response (CCyR) in one patient. Correlative translational studies suggested up-regulation of XIAP as a possible mechanism of resistance; all responses were accompanied by decreases in cIAP1 levels. However, further development of this agent in MF is uncertain.

## 7. Other Novel Targets in MF

The CD123-directed fusion protein tagraxofusp was studied in 32 patients with MF, 22 of whom had received prior JAK inhibitor therapy [96]. Patients with any degree of thrombocytopenia could enroll. Capillary leak syndrome (CLS) occurred in three patients (9%), including grades 3 and 4 in one patient each. Responses were modest, with four patients reporting CI in anemia, spleen and/or symptoms per the 2013 IWG-MRT criteria [58]. Inhibition of nuclear–cytoplasmic transport by the class of compounds known as selective inhibitors of nuclear export (SINE), e.g., selinexor and eltanexor, has been shown to lead to nuclear accumulation of p53 and induce apoptosis of *JAK2* V617F^+^ cell lines resistant to JAK inhibition, as well as synergize with ruxolitinib both in vitro and in vivo [97]. Accordingly, a phase 2 trial of selinexor in patients with MF refractory or intolerant to JAK inhibitors (ESSENTIAL) is underway (NCT03627403). The expression of programmed death 1 (PD-1) and programmed death ligand 1 (PD-L1) are increased in MPN cells [98], and immune checkpoint inhibition via antagonism of PD-1 by pembrolizumab has been demonstrated to enhance/restore mutant calreticulin-specific T-cell immunity [99]. Clinical studies of pembrolizumab (NCT03065400) and nivolumab (NCT02421354) have been conducted in patients with MF, and results are awaited. Table 3 summarizes the clinical data available thus far on the agents discussed in this review that have been studied as monotherapy in MF.

## 8. Ropeginterferon alfa-2b in PV and Beyond

Ropeginterferon alfa-2b (ropeg, formerly P1101) is a novel, long-acting, monopegylated interferon alfa-2b approved in Europe in 2019 for the treatment of PV in patients without symptomatic splenomegaly [21]. In the phase 1/2 PEGINVERA study, 51 patients with PV, a third of whom had received prior HU, were treated [101]. Ropeg was administered subcutaneously every 2 weeks in the first year, after which responding patients could switch to treatment every 4 weeks. Doses were individually optimized in the phase 2 portion (n = 26), up to the MTD of 540 µg every 2 weeks [102]. The median duration of exposure to ropeg was 5.1 years: patients were treated on the 2-weekly regimen for a median of ~2 years, and on the 4-weekly regimen for a median of ~4 years. Over 95% of the 46 evaluable patients responded, with 64.3% achieving a complete hematologic response (CHR). The time on treatment needed for 50% of patients to achieve any hematological response was ~10 weeks; the time for 50% to achieve CHR was ~1.4 years. Importantly, switching from the administration of ropeg every 2 weeks to every 4 weeks did not affect the maintenance of response. Molecular responses were achieved in >70% of patients, with 28.6% achieving a complete molecular response (CMR). The median time to any molecular response was 8 months, and that to CMR was 1.6 years.

Ropeg was then compared to HU in a phase 3, pivotal trial, PROUD-PV, the primary endpoint of which, non-inferiority of ropeg (n = 127) to HU (n = 127) in terms of CHR rate after 12 months of treatment, was met (43.1% for ropeg and 45.6% for HU) [103]. Eligible patients could be cytoreductive therapy-naïve or pre-treated with HU for <3 years as long as not in CR. After 12 months, patients on the HU arm could receive BAT in the CONTINUATION-PV portion of the study, although most (97%) remained on HU. Importantly, superiority of ropeg (n = 95) over control (n = 76) in terms of ORR (70.5% versus ~50%, *p* = 0.01) emerged after 2 and 3 years of treatment. The rate of CHR plus improvement in disease burden (splenomegaly, symptoms) at 3 years was 52.6% in the ropeg arm and 37.8% in the control arm (*p* = 0.04). Molecular responses followed the same pattern, with no difference observed at 12 months, to be followed by a highly statistically significant difference favoring ropeg (~67% versus ~30%) appearing at 2 and 3 years. Of note, allele burdens of non-driver mutations, e.g., *TET2, DNMT3A, ASXL1, EZH2*, also declined in the ropeg arm, with a statistically significant difference observed versus control at 2 years (*p* = 0.036). Ropeg was well tolerated, with cytopenias being less frequent than in the control arm, although liver enzyme elevations and myalgia were more common. The majority of patients switched from 2-weekly to 3- or 4-weekly administration of ropeg. After 4 years of treatment, the CHR rates in the ropeg and control arms were 60.6% and 43.4%, respectively (*p* = 0.02), and the rates of molecular response were 67% and 25.7%, respectively (*p* <0.0001) [104]. Thirteen patients, all in the ropeg arm, achieved a CMR. The rate of major thromboembolic events over the entire treatment period was 3.1% in both arms, with the incidence of thrombotic events being 1.4% per patient-year for ropeg and 1.2% for control. The number of patients with endocrine (3.9%), musculoskeletal (1.6%), skin (1.6%), immune system (0.8%) and psychiatric (0.8%) disorders in the ropeg arm (n = 127) remained low. This agent has also been studied in patients with pre-PMF, with improvements observed after 2 years in anemia, thrombocytosis, leukocytosis and elevated lactate dehydrogenase levels, but not in splenomegaly [105]. 

Results of a pre-planned, interim analysis of the LOW-PV trial, comparing ropeg plus phlebotomy and aspirin to phlebotomy and aspirin alone in 100 individuals with low-risk PV, were released as a late-breaking abstract at the 25th annual Congress of the EHA earlier this year [106]. The primary endpoint, the proportion of patients maintaining median hematocrit values <45% for 12 months in the absence of signs of disease progression, was achieved by 84% of patients in the ropeg group versus 60% in the phlebotomy and aspirin group (*p* = 0.008). The need for phlebotomy and symptom severity were both significantly reduced in the ropeg arm. Although AEs were significantly more common in the ropeg arm, grade 3 AEs were not. Ropeg does, therefore, have the potential to transform the management of both high-risk and low-risk PV. In the US, a clinical trial of ropeg versus anagrelide in patients with HU-resistant/intolerant ET (NCT04285086) has been announced. 

## 9. Givinostat for PV

Although clearly active, HDACis have been difficult to develop clinically for the treatment of MPNs [107], but givinostat (formerly ITF2357) could be an exception [108]. In *JAK2* V617F^+^ cell lines, as well as in primary cells from patients with PV and ET, givinostat selectively inhibited the growth and proliferation of *JAK2* V617F^+^ cells, down-regulating the mutant protein and blocking downstream signaling [109]. It also synergized with HU against *JAK2* V617F^+^ cell lines, with HU counteracting the induction of p21 typical of HDACis [110]. In a small phase 2 study in 12 PV, 1 ET and 16 MF patients, givinostat led to one CR and six PRs among the PV/ET patients; pruritus disappeared in most patients and reduction of splenomegaly was observed in 75%. Givinostat was added to the MTD of HU in 44 patients with PV unresponsive to the MTD of HU [111]. Approximately half the patients achieved CR or PR by the ELN criteria [69], and almost two-thirds achieved control of pruritus. The results of a two-part, phase 1b/2 study of givinostat in 47 patients with PV were recently published [112]. Prior HU, interferon alfa or anagrelide were permitted, but not required. Twelve patients were enrolled to the phase 1 portion (part A); the MTD of givinostat was determined to be 100 mg twice daily, and 35 patients were enrolled and treated at this dose in the phase 2 portion (part B). The ORR was 72.7% in part A and 80.6% in part B; but the vast majority of responses were PRs. Givinostat was effective in normalizing blood counts, controlling disease-related symptoms including pruritus, reducing spleen volume and the *JAK2* V617F allele burden. Diarrhea (51.4%), thrombocytopenia (45.7%) and increased serum creatinine (37.1%) were the most frequent treatment-emergent AEs. While there are plans to pursue the development of givinostat for the treatment of PV, details of the trial design and the specific patient population to be studied remain unknown at the present time. Based on the idasanutlin experience, long-term tolerability and proactive management of toxicities are of paramount importance for successful drug development in an indolent neoplasm such as PV. 

## 10. Hepcidin Mimetics for Hematocrit Control in PV 

Iron deficiency is a hallmark of PV, and is exacerbated by phlebotomy [113]. A novel therapeutic strategy, aimed at achieving superior hematocrit control, reducing/eliminating phlebotomy requirements and correcting iron deficiency, involves the use of hepcidin mimetics. NCT04057040 is a phase 2 trial of a novel hepcidin mimetic, PTG-300, administered weekly subcutaneously in phlebotomy-requiring patients with PV. Concomitant cytoreductive therapy at a stable dose is allowed. After a 28-week, open-label dose escalation phase, during which each subject’s dose is optimized, subjects are randomized 1:1 to continue PTG-300 or to receive placebo. This is followed by a one-year open-label extension phase. The primary efficacy endpoints are the proportion of subjects with the absence of phlebotomy eligibility and TSS at week 42; secondary endpoints include changes in various iron parameters from the baseline. The first results of this study are eagerly anticipated. 

## 11. Conclusions

JAK inhibitors represent a tremendous therapeutic advancement in the MPN field, having brought unprecedented benefits to patients, particularly those with MF, and their success has underscored the central role that overactive JAK-STAT signaling plays in these diseases [1]. Over the last decade, ruxolitinib has revolutionized the management of MF, and the US approval of fedratinib in 2019 has provided an important treatment option to patients after ruxolitinib failure, as well as a frontline therapeutic alternative for some patients. The widespread use of JAK inhibitors has brought the need for effective treatment of anemia further into focus, and the development of the activin receptor ligand traps has been exciting in this regard [100]. Although very early, inhibitors of ACVR1/ALK2 are also in development (NCT04455841). Beyond cytopenias, as patients with MF live longer with the disease, clinicians have learned to manage other issues associated with long-term use of ruxolitinib, such as weight gain, hyperlipidemia and infectious risks. Improvements in our understanding of the molecular events underlying disease pathogenesis and progression, particularly the role of “non-driver” mutations, have not only enabled more refined prognostication [114,115,116,117], but may also inform therapeutic choices. For example, spleen response to ruxolitinib in MF is inversely correlated with the number of non-driver mutations; patients with ≥3 mutations have a shorter time to treatment discontinuation and OS than those with fewer mutations [118]. Mutations in *IDH1*/2 are much more frequently encountered in blast phase than in chronic phase MPNs [119,120], allowing targeting by small-molecule inhibitors, providing a much-needed additional therapeutic strategy in a disease with a dismal prognosis [121]. One can already foresee a future scenario in which MF patients with more proliferative disease and robust blood counts are treated with ruxolitinib or fedratinib in combination with a BET inhibitor (e.g., CPI-0610) or BH3 mimetic (e.g., navitoclax) and those with anemia receive either momelotinib or one of the currently approved JAK inhibitors in combination with luspatercept, while those with severe thrombocytopenia at baseline are offered pacritinib. If successful in planned phase 3 trials, agents such as imetelstat or KRT-232 could fit into the treatment algorithm after JAK inhibitor failure, while anti-fibrotic agents could potentially be incorporated early, when the disease is less molecularly complex and the fibrosis more amenable to reversal. In PV, ropeginterferon alfa-2b may be poised to fundamentally alter the treatment paradigm, while the hepcidin mimetic story is intriguing. Indeed, these are exciting times for MPN patients and clinical investigators.

## Figures and Tables

**Figure 1 cancers-12-02891-f001:**
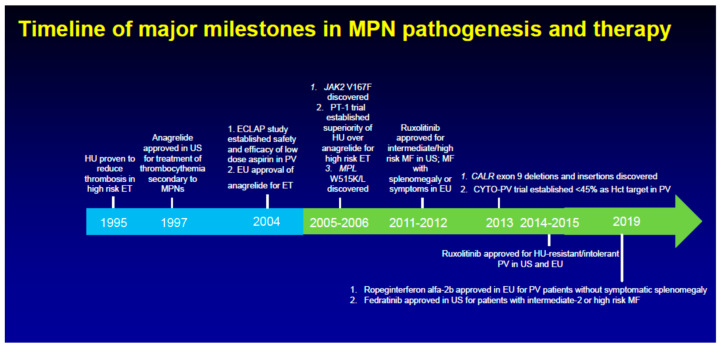
Timeline of major milestones in MPN pathogenesis and therapy.

**Figure 2 cancers-12-02891-f002:**
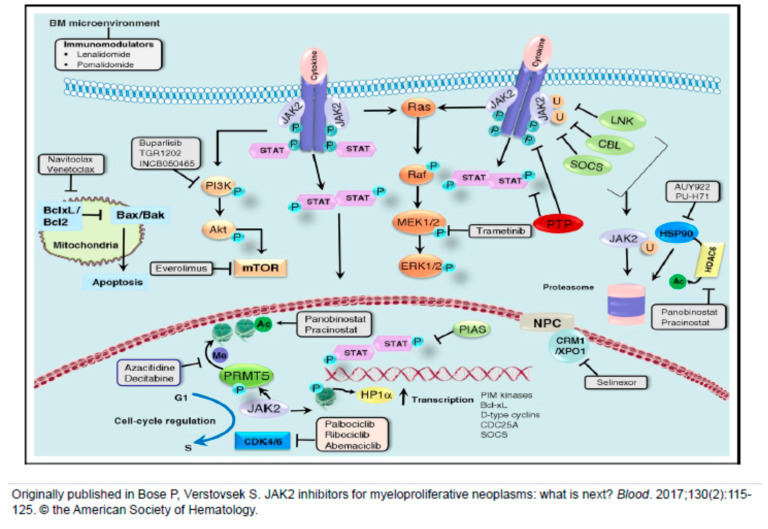
JAK2 transduces cytokine and growth factor signals from membrane-bound receptors through phosphorylation of the signal transducer and activator of transcription (STAT) family of transcription factors. Negative regulators of JAK2, such as LNK (lymphocyte adaptor protein), CBL (Casitas B-cell lymphoma) and SOCS (suppressor of cytokine signaling), lead to ubiquitinylation and proteasomal degradation of JAK2, whereas protein tyrosine phosphatases (PTPs) dephosphorylate cytokine receptors, JAKs, and STATs. The protein inhibitor of STATs (PIAS) prevents the binding of STATs to target DNA. JAK2 is a client of the chaperone protein heat shock protein 90 (HSP90), and HSP90 inhibitors and histone deacetylase 6 (HDAC6) inhibitors (through acetylation and disruption of HSP90 function) promote degradation of JAK2. JAK2 signals downstream of the PI3K/Akt/mTOR and Ras/Raf/MEK/ERK signaling cascades, which provides opportunities for combined inhibition of JAK2 and phosphatidylinositol-3-kinase (PI3K), mammalian target of rapamycin (mTOR) or mitogen activated protein kinase kinase 1/2 (MEK1/2). BH3 mimetics promote mitochondrial apoptosis, and synergism with ruxolitinib in MPN cells and animal models has been shown and validated in patients with myelofibrosis (MF). Synergism between ruxolitinib and the selective inhibitor of nuclear export (SINE) selinexor has also been demonstrated preclinically, and selinexor monotherapy is currently under study in a clinical trial in MF. Activated JAK2 promotes cell cycle progression, making combined inhibition of JAK2 and cyclin dependent kinases 4/6 (CDK4/6) a rational approach. Finally, nuclear JAK2 phosphorylates histone H3, activating transcription of many genes, including those encoding the proto-oncogene serine/threonine-protein (PIM) kinases, Bcl-xL, D-type cyclins, the cell cycle phosphatase CDC25A and SOCS (negative feedback). PIM kinase inhibitors are being studied, both alone and in combination with ruxolitinib. Epigenetic deregulation is frequent in MPNs, and combinations of ruxolitinib with epigenetic modifiers such as azacitidine and the bromodomain and extra-terminal (BET) protein inhibitor CPI-0610 have shown promise in patients with MF, as have single agents such as CPI-0610 or the lysine-specific demethylase 1 (LSD1) inhibitor, bomedemstat. Yet another epigenetic target is the arginine methyltransferase, PRMT5. Figure reproduced from Bose P, Verstovsek S. JAK2 inhibitors for myeloproliferative neoplasms: what is next? Blood. 2017, 130(2):115–125. © the American Society of Hematology.

**Table 1 cancers-12-02891-t001:** Definition of myelofibrosis (MF) relapsed/refractory to JAK inhibitor therapy in the IMBARK™ trial [40].

Documented Progressive Disease during or after JAK Inhibitor Therapy:
◦ Patients must have worsening of splenomegaly-related abdominal pain at any time after the start of JAK inhibitor therapy and EITHER: ▪ No reduction in spleen volume or size after 12 weeks of JAK inhibitor therapy, OR ▪ Worsening splenomegaly at any time after the start of JAK inhibitor therapy documented by: ➢ Increase in spleen volume from nadir by 25% measured by MRI or CT, or ➢ Increase in spleen size by palpation

Abbreviations: JAK, Janus kinase; MRI, magnetic resonance imaging; CT, computed tomography.

**Table 2 cancers-12-02891-t002:** Definition of MF relapsed/refractory to JAK inhibitor therapy in the KRT-232 trial [70].

**RELAPSED: Progressive disease any time while on ruxolitinib/JAK inhibitor**
Defined as: Increase in spleen volume by ≥25% from nadir by MRI/CT Appearance of new splenomegaly palpable ≥5 cm below LCM ≥100% increase in palpable distance below LCM for baseline splenomegaly of 5–10 cm ≥50% increase in palpable distance below LCM for baseline splenomegaly of >10 cm
**REFRACTORY:** **Lack of spleen response after ≥ 12 weeks of ruxolitinib/JAK inhibitor**
Defined as: Persistent splenomegaly, by physical exam, that is palpable ≥ 5 cm below the left LCM *AND* TSS of ≥ 10 by MPN-SAF TSS 2.0 or single symptom score ≥ 5 or two symptom scores ≥ 3, including only the symptoms of LU quadrant pain, bone pain, itching or night sweats

Abbreviations: JAK, Janus kinase; MRI, magnetic resonance imaging; CT, computed tomography; LCM, left costal margin; TSS, total symptom score; MPN-SAF, myeloproliferative neoplasm symptom assessment form; LU, left upper.

**Table 3 cancers-12-02891-t003:** Selected novel, single-agent, non-JAK inhibitor approaches in clinical trials in MF.

Agent	MOA/Drug Class (Route)	Clinicaltrials.Gov Identifier	Phase	Primary Objective or Endpoint	Results [ref.]
Imetelstat [40]	Telomerase inhibitor (IV)	NCT02426086	2 (n = 107)	≥35% SVR and ≥50% TSS reduction rates at 24 weeks; OS a key secondary endpoint	Spleen and symptom response rates at 24 weeks 10.2% (six of 59) and 32.2% (19 of 59); median OS 28.1 months (all 9.4 mg/kg arm)
PRM-151 [54]	Anti-fibrotic agent (IV)	NCT01981850	2 (n = 97)	BM fibrosis reduction by ≥1 grade at any time	27.8% (27 of 97) had BM fibrosis grade decrease at any time; 34% (32/94) had ≥50% TSS reduction; 41% (31/76) had some SVR
CPI-0610 (monotherapy) [78]	BET inhibitor (oral)	NCT02158858	2 (n = 43)	≥35% SVR rate in non-RBC TD patients; rate of TI in RBC TD patients	TD→TI conversion rate 21% (3/14 evaluable) and ≥35% SVR rate 24% (5/21 evaluable) at 24 weeks
Bomedemstat [81]	LSD1 inhibitor (oral)	NCT03136185	2 (n = 36)	Safety, PK, SVR	Most common TEAE dysgeusia (33%); no DLTs; some SVR in 83% (10/12) and TSS reduction in 86% (12/14) of evaluable pts
KRT-232 [70]	MDM2 inhibitor (oral)	NCT03662126	2 (n = 82)	≥35% SVR at week 24	Best spleen response rate 16% (four of 25 evaluable) and best symptom response rate 30% (eight of 27 evaluable) at 240 mg on d1-7 q28d
LCL-161 [95]	Smac-mimetic (oral)	NCT02098161	2 (n = 50)	ORR by IWG-MRT 2013 criteria	30% ORR (15 of 50 pts); six anemia responders
Tagraxofusp [96]	CD123-directed fusion protein (IV)	NCT02268253	1/2 (n = 32)	Determination of RP2D; defining efficacy and safety	CLS occurred in three pts (9%); IWG-MRT 2013 spleen response in one pt; 56% (10 of 18 evaluable) had some spleen size reduction; 46% (11 of 24 evaluable) had some symptom improvement
Alisertib [57]	Aurora kinase inhibitor (oral)	NCT02530619	Pilot (n = 24)	Safety; response assessment by IWG-MRT 2013 criteria	Most frequent TEAEs cytopenias, N/V/D, mucositis; response rates in evaluable pts: spleen 29% (4/14), symptoms 32% (7/22), anemia 11% (2/19)
Luspatercept (monotherapy) [100]	Activin receptor ligand trap (subcutaneous)	NCT03194542	2 (n = 43)	Anemia response: Hgb improvement or RBC TI	14% (n = 22) had Hgb improvement and 10% (n = 21) achieved RBC TI (with monotherapy)

Abbreviations: SVR, spleen volume reduction; OS, overall survival; BM, bone marrow; TD, transfusion-dependent; RBC, red blood cell; TI, transfusion-independent; PK, pharmacokinetic; TEAE, treatment-emergent adverse event; DLT, dose limiting toxicity; ORR, overall response rate; IWG-MRT, International Working Group for Myelproliferative Neoplasm Research and Treatment; RP2D, recommended phase 2 dose; CLS, capillary leak syndrome; N/V/D, nausea, vomiting, diarrhea; Hgb, hemoglobin.

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
