# Peer review of "Novel Concepts of Treatment for Patients with Myelofibrosis and Related Neoplasms"

_cancers, 2020, doi:10.3390/cancers12102891_

Round 1

Reviewer 1 Report

The review by Bose et al. provides a complete overview of the new possibilities of treatment of myeloproliferative neoplasms starting from the benefit of JAK2 inhibition and giving a reasoned description of the therapies that have been developed to overcome resistance to conventional therapies. The authors have focused on the possible approaches for targeting various pathways such as those regulating telomerase activity, bone marrow fibrosis, p53-dependent apoptosis, as well as the anti-apoptotic machinery, the chromatin remodeling, and others.

The manuscript is very well written, the discussion of the various topics is very clear and the citation of the most relevant literature, even very recent, is accurate.

Minor points:

The conclusions lack real take-home massages. For example, a discussion on possible decision algorithms could be proposed.

A graphic timeline on the development of new treatments from HU to the latest and a figure on the pathways that can be targeted in the treatment of MPN could be useful.

Author Response

We thank the reviewer for his/her very favorable review of our manuscript. The 2 minor points raised are addressed below.

  1. The conclusions section has been expanded to include a brief discussion on possible treatment algorithms, as requested (lines 437 through 444 IN THE TRACKED CHANGES VERSION).
  2. A graphic timeline of the development of various treatments for the classic MPNs has been provided (Figure 1), as well as a figure depicting pathways that can be therapeutically targeted (Figure 2).

Reviewer 2 Report

Manuscript by Bose et al highlights the treatment strategies being used in clinical trials with jak inhibitors in myelofibrosis patients and summarizes the novel approaches with or without commonly used ruxolitinib. The review is written well in most of the part.

Minor comments

Can authors highlight navitoclax in patinets who are intolerant to jak inhibitors? Would ABT 737 combination with ruxolitinib worsen the thrombocytopenia?

Author Response

We thank the reviewer for his/her overall favorable impression of our manuscript. 

We have now addressed the reviewer's minor comments in lines 277 through 289 IN THE TRACKED CHANGES VERSION.

Reviewer 3 Report

The authors have made an impressive and extensive literature study regarding the current treatment strategies on MPN diseases. They provide with lots of information about past and on-going clinical studies on many different agents targeting mainly the JAK-STAT pathway.

Given the many different outcomes of mono- or combined agent treatments that the authors conclude that will/should be further investigated there is limited discussion on the genomic landscape (especially the non-driver mutations) of patients that could possibly affect the results of the treatments. In addition limited literature has been included on this direction. There are actually quite a few studies published in big cohorts of patients.

Also, it would be very interesting to elaborate a bit more the conclusions section with facts from the described clinical studies explaining how in the last decade the developments on the discovery on new agents has impacted the management of MPN patients in the clinic and how the new genomic era at the single cell level could contribute to the development of targeted treatments with less side effects.

Author Response

We thank the reviewer for his/her comments. In response to the specific suggestions for improvement, we have expanded the conclusions section to address the points raised by the reviewer from lines 411 through 431 IN THE TRACKED CHANGES VERSION. 

Reviewer 4 Report

Bose and colleagues present a comprehensive review of novel concepts of treatment for patients with myelofibrosis (MF) and related neoplasms. The topic is of relevance to the field. The manuscript is a well-written review of the current literature and focused on agents tested as monotherapy in MF patients after JAK inhibitor resistance, and on new compounds that have shown efficacy in MF or polycythemia vera (PV).

The collection of evidence is broad enough and well structured. I find this is a useful review, as it describes new agents that exploit the therapeutic vulnerabilities of these diseases other than the JAK-STAT pathway. I have no major criticisms. I would only mention two minor aspects that, in my opinion, could be improved for the sake of clarity:

1/ Table 1 and 2 are quite informative, the design is however a bit confusing (might be due to the PDF conversion)

2/ A recapitulative table with the clinical trials assessing these new agents in patients with MF or MF-related conditions would be useful.

(Agent/Mechanism-target/trial identifier/Phase/Primary Efficacy Endpoint if applicable/ Results (publication or meeting).

Minor: line 7, "Janus" not in bold.

Author Response

We would like to thank the reviewer for his/her favorable review. In response to the specific points raised:

  1. Yes, this is due to the formatting changes to the paper made by the editorial office. They might wish to make further changes to improve the clarity of the tables. We have bolded a subheading ("REFRACTORY") in Table 2 to make it easier to read and understand.
  2. This has now been provided (Table 3).
  3. "Janus" is not in bold font in the first line of the abstract (now line 18). It might have appeared that way to the reviewer because of formatting issues.